# Introduction of SGLT2 Inhibitors and Variations in Other Disease-Modifying Drugs in Heart Failure Patients: A Single-Centre Real-World Experience

Erika Tabella [1], Michele Correale [2], Gianmarco Alcidi [1], Rosanna Pugliese [1], Sara Ioannoni [1], Matteo Romano [1], Gianpaolo Palmieri [1], Matteo Di Biase [1], Natale Daniele Brunetti [1,2] and Massimo Iacoviello [1,2,*]

[1] Department of Medical and Surgical Sciences, University of Foggia, 71122 Foggia, Italy; erika.tabella@unifg.it (E.T.); gianmarco.alcidi@gmail.com (G.A.); rosanna.pug@gmail.com (R.P.); sara.ioannoni@unifg.it (S.I.); matteo.romano@unifg.it (M.R.); gianpaolo.palmieri@libero.it (G.P.); dibiama@gmail.com (M.D.B.); natale.brunetti@unifg.it (N.D.B.)

[2] Cardiology Unit, Polyclinic University Hospital of Foggia, 71122 Foggia, Italy; michele.correale@libero.it

\* Correspondence: massimo.iacoviello@gmail.com

**Abstract:** Background: The sodium–glucose cotransporter-2 inhibitors (SGLT2i) have emerged as a crucial therapeutic option for patients with chronic heart failure with reduced ejection fraction (HFrEF). The aim of this study was to evaluate, in a real-world population from a single centre, the feasibility of introducing SGLT2i and their interaction with other recommended drug classes. Methods: Consecutive patients affected by chronic heart failure (CHF) were evaluated beginning in January 2022. At the baseline clinical visit, both the patient's current medication and the prescribed treatments were recorded. Over a 6- to 12-month follow-up, changes in concomitant therapy were analysed. Results: At baseline, among 350 patients evaluated, only 17 (5%) were already taking SGLT2i: 13 with HFrEF, five with mildly reduced (HFmrEF), preserved (HFpEF) or improved (HFimpEF) ejection fraction. After the baseline assessment, SGLT2i were prescribed to 224 (64%) of the patients, including 179 (84%) with HFrEF, 27 (42%) with HFmrEF/HFimpEF, and 18 (22%) with HFpEF/HFimpEF. After follow-up, SGLT2i therapy was well tolerated and was associated with a significant increase in sacubitril/valsartan prescriptions and a decrease in diuretic use. Finally, a significant improvement in functional status and left ventricular systolic function after SGLT2i therapy was observed. Conclusions: In this single-centre, real-world study, SGLT2i were primarily prescribed to HFrEF patients who were already on other recommended drug classes for their treatment. Additionally, there was a noticeable enhancement in the prescribed therapy during a short-term follow-up. These findings further bolster the inclusion of this therapeutic approach in regular clinical practice.

**Keywords:** heart failure; therapy; type-2 sodium–glucose cotransporters inhibitors; angiotensin receptor neprylisin inhibitors



## 1. Introduction

Sodium–glucose cotransporter-2 inhibitors (SGLT2i) represent a cornerstone in the treatment of patients with heart failure with reduced ejection fraction (HFrEF), as well as those with mildly reduced (HFmrEF), preserved (HFpEF), and improved (HFimpEF) ejection fraction [1–9]. The beneficial effects across the entire spectrum of left ventricular ejection fraction (LVEF) are attributed to several hypothesised mechanisms that are not yet well clarified [10,11]. Certainly, the effects of SGLT2i in terms of reduction of glomerular hyperfiltration and preservation of glomerular filtration rate may play a pivotal role in cardiorenal protection [2,11–13]. In addition to these effects, other potential direct and indirect cardiac effects on cardiac function have been hypothesised, such as diuretic effects [14], improvement in myocardial energetics [15,16], reduction of cytosolic sodium and calcium

levels and an increase in mitochondrial calcium [17]. Moreover, the increased cardiac delivery of oxygen due to the elevated hematocrit and the reduction of afterload could potentially enhance ventricular function [18]. On the basis of these hypotheses, it is likely to argue that these mechanisms are additive to those of the classes of drugs able to modulate neurohormonal activation [8,10]. Consequently, in HFrEF patients, in order to improve survival, current guidelines advise the prompt introduction of four drug classes: SGLT2i, beta-blockers, mineralocorticoid receptor antagonists (MRAs), and angiotensin receptor–neprilysin inhibitors (ARNi) or, if ARNi therapy is not tolerated, angiotensin-converting enzyme inhibitors (ACEi) or angiotensin receptor blockers (ARBs) are recommended instead [4]. In this new complex therapeutic landscape, limited data exist regarding the feasibility of these recommendations and potential interactions between some drug classes, such as SGLT2i and ARNi [19,20].

Furthermore, the decision to prescribe SGLT2i depends not only on clinical evidence and guideline recommendations but also on the stipulations set by regulatory bodies and the reimbursement policies of different national healthcare systems. For instance, in Italy, reimbursement for SGLT2i was restricted to patients with type 2 diabetes mellitus (T2DM) until February 2022. Only subsequently could dapagliflozin, followed by empagliflozin, be prescribed with reimbursement for patients with HFrEF. Furthermore, until June 2023, reimbursement was not permitted for patients with HFmrEF and HFpEF, despite existing research supporting their efficacy [4,5].

Given these considerations, this study aimed to assess, in a real-world setting from a single centre, the feasibility of introducing SGLT2i and their interactions with other recommended drug classes.

## 2. Materials and Methods

We evaluated patients referred to the Heart Failure Unit of the University Policlinic Hospital of Foggia for the diagnosis of chronic heart failure (CHF) beginning in February 2022 when the reimbursement of SGLT2i was allowed by the Italian National Health System. For the study, all the patients with a history of CHF were considered independently from LVEF, NYHA class, and eligibility for SGLT2i therapy. Concerning SGLT2i eligibility, following the indications of the Italian Ministry of Health, dapagliflozin (from February 2022) and empagliflozin (from June 2022) could be prescribed and reimbursed for patients in the New York Heart Association (NYHA) class II–III with an LVEF ≤40%. Additionally, regardless of their LVEF, all SGLT2i prescriptions could be reimbursed for CHF patients also diagnosed with T2DM. All patients were enrolled and included in the Daunia registry, which is also aimed to study the effects of novel therapeutic approaches on clinical outcomes. This registry has received approval from local ethics committees, and all participating patients provided written informed consent.

Baseline evaluations. The baseline evaluation was considered the first recorded medical visit after February 2022. During this visit, patients underwent a physical examination, a 12-lead electrocardiogram, as well as one- and two-dimensional echocardiographic evaluations. Peripheral blood samples were also collected. Medical records noted the presence of conditions such as ischaemic cardiomyopathy, cerebrovascular disease or stroke, arterial hypertension, atrial fibrillation, diabetes mellitus, and dyslipidaemia. Additionally, HF status, NYHA class, and antidiabetic therapy were recorded. Echocardiographic assessments utilised a phased-array echo-Doppler system (EPIQ CVx system, Philips, Amsterdam, the Netherlands) to evaluate LVEF using the Simpson method. Based on LVEF values, patients were classified as having HFrEF if their LVEF was <40%. The remaining patients were classified into the categories of HFmrEF, HFpEF, and HFimpEF in line with the current universal definition of HF [21]. Creatinine serum concentrations (mg/dL) were measured at the baseline evaluation. Subsequently, the glomerular filtration rate (GFR) (mL/min) was calculated using the Chronic Kidney Disease Epidemiology Collaboration (CKD-EPI) formula [22]. Medication dosages for HF were standardised as it is described in the following. ACEi doses were converted to the equivalent enalapril dose. Specif-

ically, enalapril 20 mg/die is equivalent to ramipril 10 mg/die, zofenopril 30 mg/die, and lisinopril 20 mg/die. For ARBs, doses were translated to the valsartan equivalent: valsartan 320 mg/die corresponds to losartan 100 mg/die or candesartan 32 mg/die [4]. Beta-blocker doses were standardised to the bisoprolol equivalent: bisoprolol 10 mg/die equates to carvedilol 50 mg/die, nebivolol 10 mg/die, or metoprolol tartrate 200 mg/die. Lastly, for sacubitril/valsartan, 24/26 mg bid is equivalent to 100 mg/die, 49/51 mg bid to 200 mg/die, and 97/103 mg bid to 400 mg/die.

Follow-up. Patients underwent regular check-ups based on the protocol of our outpatient HF clinic, ensuring a minimum of one assessment every 6 months. Although, at baseline, HfrEF patients were already being treated with ARNi (sacubitril/valsartan), ACEi, or ARBs (unless they were contraindicated or not tolerated) in conjunction with MRAs and beta-blockers [4], further efforts were made during follow-up to introduce and uptitrate the recommended disease modifiers drugs [4]. The minimum dosage of loop diuretic was used in order to keep patients stable. During follow-up, the dose reduction of withdrawal of loop diuretic was considered in the following cases: the presence of hypotension related to dehydration due to an excessive diuretic dose and significant clinical and/or functional and/or echocardiographic improvement. Furthermore, SGLT2i administration was initiated in T2DM-diagnosed patients who exhibited an LVEF > 40%. Each patient's 6- or 12-month data were scrutinised to observe alterations in the parameters under study and their therapeutic regimen.

Statistical analysis. The representation of continuous data was in the form of mean values ± standard deviations. Discrete variables were summarised as frequencies and percentages. The Student's t-test and Fisher's exact test were employed to discern differences between patients with and without LVEF < 40%. For gauging parameter shifts among patients on SGLT2i therapy, we used the Student's t-test for paired samples and McNemar's test for continuous and categorical data, respectively. Analyses were performed using Statistica 6.1 software (StatSoft Inc., Tulsa, Oklahoma). A *p*-value of < 0.05 was considered statistically significant.

## 3. Results

Of the 350 patients assessed since the commencement of the indexed period, 213 (61%) showed an LVEF $\leq$ 40% (HfrEF) and 137 (39%) showed >40%. Among the latter, 70 (20% of all patients) were classifiable as HfimpEF, 27 (8% of all patients) with LVEF between 41 and 49% as HfmrEF, and 38 (11%) with LVEF > 50% as HfpEF. Table 1 shows the baseline clinical characteristics of all enrolled patients, as well as those with and without HfrEF. HfrEF patients predominantly belonged to the male demographic, had an ischemic aetiology, were more frequently treated with beta-blockers and ARNi, and had a cardioverter defibrillator with or without cardiac resynchronisation therapy (CRT). Furthermore, they exhibited a reduced incidence of hypertension, a diminished baseline systolic blood pressure, and a more advanced NYHA class. Table 1 shows the comparisons among HfrEF, HfmrEF, and HfpEF. Patients with HfpEF were less frequently males and with ischemic aetiology, with less functional limitation, and higher systolic arterial pressure. Among patients with HfmrEF and HfpEF, a relevant proportion had initially been diagnosed with HfrEF. This can explain the percentage of patients taking ARNi and carrying ICD/CRT.

**Table 1.** Patient baseline clinical characteristics.

| | All Patients | LVEF < 40% (HfrEF) | LVEF 41–49% (HfmrEF or HfimpEF) | LVEF $\geq$ 50% HfpEF or HfimpEF | *p* |
|---|---|---|---|---|---|
| Number | 350 | 213 | 59 | 78 | |
| Age (years) | 66 ± 12 | 65 ± 12 | 66 ± 11 | 67 ± 13 | 0.448 |
| Males, (%) | 280 (80) | 182 (85) | 48 (81) | 50 (64) | <0.001 |

**Table 1.** *Cont.*

| | All Patients | LVEF < 40% (HfrEF) | LVEF 41–49% (HfmrEF or HfimpEF) | LVEF ≥ 50% HfpEF or HfimpEF | *p* |
|---|---|---|---|---|---|
| De novo HF, n (%) | 16 (5) | 13 (6) | 3 (5) | 0 (0) | 0.086 |
| HfimpEF, n (%) | 70 (20) | - | 32 ((54) | 38 (49) | - |
| Ischemic aetiology, n (%) | 147 (42) | 105 (49) | 22 (37) | 20 (26) | 0.001 |
| Diabetes mellitus, n (%) | 126 (36) | 75 (35) | 22 (37) | 29 (37) | 0.929 |
| Arterial hypertension, n (%) | 230 (66) | 128 (60) | 46 (78) | 56 (72) | 0.019 |
| Atrial fibrillation, n, (%) | 48 (14) | 27 (13) | 7 (12) | 14 (18) | 0.462 |
| NYHA class I, n (%) | 27 (7.7) | 10 (4.7) | 4 (6.8) | 13 (16.7) | |
| II, n (%) | 193 (55.1) | 124 (58.2) | 31 (52.5) | 38 (48.7) | 0.017 |
| III, n (%) | 130 (37.2) | 79 (37.1) | 27 (34) | 27 (34.6) | |
| SAP (mm Hg) | 124 ± 19 | 121 ± 18 | 127 ± 22 | 132 ± 18 | 0.043 |
| Heart rate (beats/minute) | 68 ± 12 | 68 ± 12 | 67 ± 9 | 70 ± 16 | 0.356 |
| LVEF (%) | 39 ± 9 | 32 ± 6 | 45 ± 2 | 53 ± 3 | <0.001 |
| Creatinine (mg/dl) | 1.30 ± 0.8 | 1.30 ± 0.6 | 1.17 ± 0.6 | 1.39 ± 1.2 | 0.038 |
| GFR-EPI (mL/min/1.73 m$^2$) | 62.5 ± 24.7 | 61 ± 21 | 71 ± 25 | 61 ± 28 | 0.053 |
| Concomitant therapy at the enrollment | | | | | |
| ARNi, n (%) | 179 (51) | 139 (51) | 23 (39) | 17 (22) | <0.001 |
| Sacubitril/valsartan dose (mg/die) | 191 ± 134 | 191 ± 124 | 254 ± 137 | 276 ± 139 | 0.006 |
| ACE-I, n (%) | 66 (19) | 33 (15) | 15 (25) | 20 (26) | 0.067 |
| Enalapril equivalent dose (mg/die) | 9.9 ± 6.7 | 9.1 ± 6.7 | 9.6 ± 6.3 | 11.6 ± 7.1 | 0.449 |
| ARB, n (%) | 52 (15) | 20 (9) | 10 (17) | 22 (28) | <0.001 |
| Valsartan equivalent dose (mg/die) | 139 ± 103 | 53 ± 49 | 73 ± 55 | 109 ± 91 | 0.047 |
| Beta-blockers, n (%) | 330 (94) | 207 (97) | 53 (90) | 70 (90) | 0.015 |
| Bisoprolol equivalent dose (mg/die) | 4.8 ± 3.2 | 4.9 ± 3.3 | 4.3 ± 2.6 | 4.8 ± 3.2 | 0.369 |
| MRA, n (%) | 235 (67) | 145 (68) | 39 (66) | 845 (58) | 0.255 |
| MRA dose | 38.4 ± 30.1 | 39.1 ± 29.2 | 34.8 ± 28.2 | 39.2 ± 33.9 | 0.674 |
| Loop diuretics, n (%) | 250 (71) | 156 (73) | 41 (69) | 53 (68) | 0.633 |
| Furosemide equivalent dose (mg/die) | 71 ± 86 | 49 ± 70 | 55 ± 93 | 55 ± 93 | 0.807 |
| ICD and/or CRT, n (%) | 191 (55) | 139 (65) | 26 (44) | 23 (29) | <0.001 |
| SGLT2i | | | | | |
| Before baseline evaluation, n (%) | 17 (5) | 11 (5) | 3 (5) | 3 (4) | |
| After baseline evaluation, n (%) | 207 (59) | 168 (79) | 24 (41) | 15 (19) | <0.001 |

*p* refers to ANOVA or Pearson's Chi-square according to the analysis of continuous or categorical variables for the three analysed subgroups. ACE-I: inhibitors of angiotensin-converting enzyme; ARB: angiotensin II receptor blockers; ARNi: angiotensin receptor neprilysin inhibitors; GFR-EPI: estimated glomerular filtration rate by EPI formula; CRT: cardiac resynchronisation therapy; ICD: implantable cardioverter defibrillator; HfimpEF: heart failure with improved ejection fraction; HfmrEF: heart failure with mildly reduced ejection fraction; HfpEF: heart failure with preserved ejection fraction; HfrEF: heart failure with reduced left ventricular ejection fraction; LVEF: left ventricular ejection fraction; MRA: mineralocorticoid receptor antagonists; NYHA class: New York heart Association class; SAP: systolic arterial pressure.

### 3.1. SGLT2 Inhibitor Therapy

At baseline, only 17 (5%) patients were already taking SGLT2i: 11 with HfrEF, three with HfmrEF, and three with HfpEF. After baseline evaluation, SGLT2i was prescribed to 224 (64%) patients. This encompassed 179 (84%) with HfrEF, 27 (46%) with HfmrEF/HfimpEF, and 18 (23%) with HfpEF/HfimpEF. Dapagliflozin was prescribed to 187 (83%) patients, empagliflozin to 35 (16%), and canagliflozin to 2 (1%). Notably, canagliflozin was the preferred choice by diabetologists before the study's onset for diabetes-afflicted patients with an LVEF > 40%, coupled with compromised renal function and proteinuria. Patients with HfmrEF/HfpEF/HfimpEF less frequently received a prescription of SGLT2i. Patients with HfmrEF/HfpEF/HfimpEF were more frequently diabetic than HfrEF (67% vs. 31%).

After 6 to 12 months of follow-up, only 7 of 224 patients discontinued SGLT2i due to hypotension (2), acute kidney injury (1), intolerance (2), and urinary tract infection (2). Figure 1 shows the main classes of drugs prescribed at baseline.

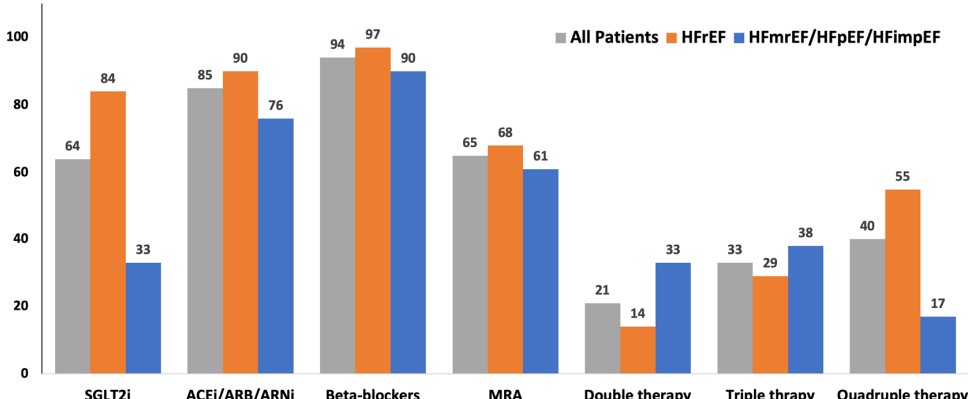

**Figure 1.** Therapy prescribed at baseline. The main classes of drugs prescribed at baseline in all the enrolled patients, in those with HfrEF, and those with HfmrEF or HfpEF. ACE: angiotensin-converting enzyme; ARB: angiotensin II receptor blockers; ARNi: angiotensin II receptor and neprylisin inhibitor; HfmrEF: heart failure with mild reduced left ventricular ejection fraction; HfmrEF: heart failure with preserved left ventricular ejection fraction; HfrEF: heart failure with reduced left ventricular ejection fraction; MRA: mineralocorticoid receptor antagonists; SGLT2i: inhibitors of type 2 sodium–glucose cotransporter.

In Table 2, the clinical characteristics of patients receiving empagliflozin and dapagliflozin are shown separately for the group of HfrEF and HfmrEF/HfpEF/HfimpEF patients.

**Table 2.** Comparison between patients in whom dapagliflozin and empagliflozin were prescribed, according to left ventricular ejection fraction.

|  | Patients with HfrEF | | | Patients with HfmrEF/HfpEF/HfimpEF | | |
|---|---|---|---|---|---|---|
|  | Dapa | Empa | *p* | Dapa | Empa | *p* |
|  | n: 158 | n: 21 |  | n: 29 | n: 14 |  |
| Age (years) | 64 ± 11 | 68 ± 10 | 0.161 | 66 ± 11 | 69 ± 10 | 0.309 |
| Males (%) | 84 | 95 | 0.461 | 89 | 64 | 0.149 |
| Weight (kg) | 79 ± 16 | 82 ± 23 | 0.583 | 85 ± 14 | 88 ± 19 | 0.607 |
| NYHA class | 234 ± 0.5 | 2.5 ± 0.6 | 0.087 | 2.3 ± 0.7 | 2.6 ± 0.5 | 0.103 |
| SAP (mmHg) | 120 ± 16 | 121 ± 18 | 0.621 | 122 ± 17 | 135 ± 30 | 0.072 |
| Heart rate (bpm) | 68 ± 11 | 68 ± 8 | 0.940 | 67 ± 11 | 70 ± 14 | 0.481 |
| LVEF (%) | 32 ± 6 | 31 ± 6 | 0.601 | 47 ± 5 | 49 ± 4 | 0.109 |
| Creatinine (mg/dl) | 1.28 ± 0.59 | 1.27 ± 0.26 | 0.910 | 1.09 ± 0.33 | 1.31 ± 0.59 | 0.181 |
| GFR-EPI (mL/min/1.73 m$^2$) | 62 ± 21 | 59 ± 18 | 0.611 | 73 ± 25 | 69 ± 29 | 0.181 |
| Concomitant therapy at baseline |  |  |  |  |  |  |
| ARNi, % | 69 | 57 | 0.276 | 34 | 14 | 0.166 |
| Sacubitril/valsartan dose (mg/die) | 185 ± 122 | 242 ± 143 | 0.141 | 165 ± 131 | 400 ± 0 | - * |
| ACE-I, % | 15 | 14 | 0.974 | 21 | 21 | 0.955 |
| Enalapril equivalent dose (mg/die) | 8.9 ± 6.7 | 10.4 ± 9.4 | 0.731 | 8.0 ± 7.4 | 20 ± 0 | - † |
| ARB % | 54 | 41 | 0.493 | 27 | 29 | 0.946 |
| Valsartan equivalent dose (mg/die) | 54 ± 52 | 41 ± 38 | 0.688 | 150 ± 108 | 140 ± 133 | 0.891 |
| Beta-blockers (%) | 97 | 100 | 0.461 | 100 | 86 | 0.037 |
| Bisoprolol equivalent dose (mg/die) | 5.1 ± 3.4 | 4.2 ± 2.8 | 0.248 | 4.7 ± 3.1 | 5.2 ± 3.7 | 0.627 |
| MRA % | 75 | 57 | 0.090 | 83 | 64 | 0.179 |
| MRA dose (mg/die) | 42 ± 28 | 50 ± 32 | 0.826 | 43 ± 26 | 50 ± 28 | 0.466 |
| Loop diuretics % | 73 | 100 | 0.006 | 66 | 71 | 0.698 |
| Furosemide equivalent dose (mg/die) | 66 ± 77 | 59 ± 57 | 0.707 | 88 ± 114 | 78 ± 72 | 0.804 |

Data expressed as mean ± standard deviation. P refers to Student's t-test or Pearson's Chi-squared test as appropriate. * only 3 patients taking ACEi; † only 2 patients taking ARNi. ACE-I: inhibitors of angiotensin-converting enzyme; ARB: angiotensin II receptor blockers; ARNi: angiotensin receptor neprilysin inhibitors; Dapa: dapagliflozin; Empa: empagliflozin; GFR-EPI: estimated glomerular filtration rate by EPI formula; HFimpEF: heart failure with improved ejection fraction; HFmrEF: heart failure with mildly reduced ejection fraction; HFpEF: heart failure with preserved ejection fraction; HFrEF: heart failure with reduced left ventricular ejection fraction; LVEF: left ventricular ejection fraction; MRA: mineralocorticoid receptor antagonists; NYHA class: New York Heart Association class; SAP: systolic arterial pressure.

*3.2. Changes in Patients with SGLT2 Inhibitor Therapy*

As depicted in Table 3, among all patients to whom SGLT2i was prescribed at baseline, there was a significant reduction in weight, systolic arterial pressure, and NYHA class, as well as a minor but notable increase in serum creatinine levels. Furthermore, a significantly larger proportion of patients were taking ARNi during the follow-up. Concurrently, there was a significant drop in the percentage of patients on ACEi/ARBs and diuretics. No discernible differences emerged in the administration of beta-blockers and MRAs.

**Table 3.** Changes of studied parameters in patients treated with SGLT2i.

| All Patients with SGLT2i | Baseline | After | $p$ |
|---|---|---|---|
| Weight (kg) | $80.7 \pm 16.7$ | $79.6 \pm 16.4$ | 0.002 |
| NYHA class | $2.4 \pm 0.6$ | $2.3 \pm 0.6$ | 0.010 |
| SAP (mmHg) | $122 \pm 18$ | $118 \pm 18$ | 0.005 |
| Heart rate (bpm) | $68 \pm 11$ | $67 \pm 10$ | 0.463 |
| LVEF (%) | $35 \pm 8$ | $37 \pm 9$ | <0.001 |
| Creatinine (mg/dl) | $1.26 \pm 0.37$ | $1.30 \pm 0.45$ | 0.044 |
| GFR-EPI (mL/min/1.73 m$^2$) | $61 \pm 20$ | $61 \pm 21$ | 0.469 |
| Concomitant therapy | | | |
| ARNi, % | 61 | 68 | <0.001 |
| Sacubitril/valsartan dose (mg/die) | $178 \pm 133$ | $207 \pm 130$ | <0001 |
| ACE-I, % | 14 | 11 | 0.070 |
| Enalapril equivalent dose (mg/die) | $9.3 \pm 7.5$ | $8.4 \pm 6.6$ | 0.213 |
| ARB, % | 14 | 12 | 0.343 |
| Valsartan equivalent dose (mg/die) | $72 \pm 72$ | $72 \pm 76$ | 1.00 |
| Beta-blockers, % | 97 | 97 | 1.00 |
| Bisoprolol equivalent dose (mg/die) | $5.1 \pm 3.3$ | $5.3 \pm 3.3$ | 0.094 |
| MRA, % | 77 | 79 | 0.522 |
| MRA dose (mg/die) | $43 \pm 27$ | $43 \pm 25$ | 0.826 |
| Loop diuretics, % | 75 | 69 | 0.014 |
| Furosemide equivalent dose (mg/die) | $69 \pm 79$ | $63 \pm 106$ | 0.309 |
| **Patients with HFrEF and SGLT2i (n: 178)** | **Baseline** | **After** | |
| Weight (kg) | $79.5 \pm 16.8$ | $78.4 \pm 16.4$ | 0.007 |
| NYHA class | $2.3 \pm 0.6$ | $2.2 \pm 0.6$ | 0.004 |
| SAP (mmHg) | $119 \pm 16$ | $116 \pm 16$ | 0.019 |
| Heart rate (bpm) | $68 \pm 11$ | $67 \pm 10$ | 0.117 |
| LVEF (%) | $32 \pm 6$ | $35 \pm 8$ | <0.001 |
| Creatinine (mg/dl) | $1.28 \pm 0.35$ | $1.30 \pm 0.43$ | 0.183 |
| GFR-EPI (mL/min/1.73 m$^2$) | $60 \pm 18$ | $60 \pm 21$ | 0.798 |
| Concomitant therapy | | | |
| ARNi, % | 68 | 77 | <0.001 |
| Sacubitril/valsartan dose (mg/die) | $175 \pm 131$ | $206 \pm 128$ | <0001 |
| ACE-I, % | 14 | 9 | 0.027 |
| Enalapril equivalent dose (mg/die) | $7.7 \pm 6.4$ | $6.7 \pm 5.4$ | 0.189 |
| ARB, % | 10 | 8 | 0.289 |
| Valsartan equivalent dose (mg/die) | $70 \pm 35$ | $67 \pm 20$ | 0.674 |
| Beta-blockers, % | 98 | 98 | 1.00 |
| Bisoprolol equivalent dose (mg/die) | $5.1 \pm 3.3$ | $5.2 \pm 3.2$ | 0.240 |
| MRA, % | 76 | 79 | 0.359 |
| MRA dose (mg/die) | $42 \pm 28$ | $41 \pm 24$ | 0.651 |
| Loop diuretics, % | 76 | 69 | 0.009 |
| Furosemide equivalent dose (mg/die) | $63 \pm 73$ | $59 \pm 102$ | 0.342 |

Data expressed as mean $\pm$ standard deviation. $p$ refers to Student's t-test or McNemar test as appropriate. ACE-I: inhibitors of angiotensin-converting enzyme; ARB: angiotensin II receptor blockers; ARNi: angiotensin receptor neprilysin inhibitors; GFR-EPI: estimated glomerular filtration rate by EPI formula; HFrEF: heart failure with reduced left ventricular ejection fraction; LVEF: left ventricular ejection fraction; MRA: mineralocorticoid receptor antagonists; NYHA class: New York Heart Association class; SAP: systolic arterial pressure.

The second part of Table 3 separately reports the data relative to HFrEF, in whom an improvement in the adherence to the recommended use of disease modifiers' drugs was

observed. Figure 2 illustrates variations in the proportion of either not taking or being on low, average, or high doses of ARNi, beta-blockers, and MRAs among HFrEF patients in SGLT2i therapy. The proportion of patients on higher doses either remained stable or displayed a trend toward increasing.

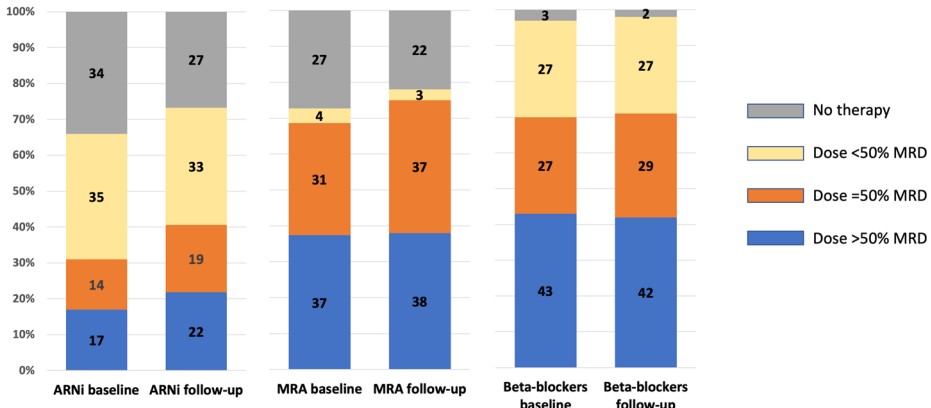

**Figure 2.** Presence and dose at baseline and during follow-up of ARNi, beta-blockers, and MRA in HFrEF patients with SGLT2i therapy. When the class of drugs is prescribed, the dose is expressed as below, equal to or above 50% of the current recommended maximum dose. ARNi: angiotensin II receptor and neprylisin inhibitor; MRA: mineralocorticoid receptor antagonists; SGLT2i: inhibitors of type 2 sodium–glucose cotransporter.

## 4. Discussion

In our single-centre, real-world study, SGLT2i therapy was prescribed to a substantial percentage of patients with HFrEF. The adoption of this therapy correlated with enhanced utilisation of other disease-modifying drugs presently endorsed for HFrEF [4]. These results hold significance for several reasons. Initially, dapagliflozin [1], followed by empagliflozin [2], proved effective in decreasing the combined endpoint of hospitalisation for HF and cardiovascular death in HFrEF. Furthermore, in the case of dapagliflozin, a significant reduction in cardiovascular mortality and overall mortality was observed [1]. Given this evidence, both the recent European [4] and American [23] guidelines recommend the use of this class of drugs with a Class I recommendation and a Level of Evidence A. Contrasting earlier stepwise methodologies, the immediate adoption of four drug classes—primarily ARNi over ACEi/ARBs, followed by MRAs, beta-blockers, and SGLT2i—that can alter the course of HFrEF is now recommended [4,23]. This novel approach is supported by recent studies, further demonstrating its beneficial effects [24,25]. However, both randomised controlled trials [1–3] and real-world data evaluating SGLT2i show a limited percentage of patients using ARNi (specifically sacubitril/valsartan). In this context, our study is significant as it reveals that not only can a vast majority of patients be introduced to SGLT2i, but their introduction also correlates with a high prevalence of sacubitril/valsartan treatment. Moreover, during the follow-up, the prescription rate of sacubitril/valsartan increased among HFrEF patients. Finally, at the end of the follow-up, among the HFrEF patients taking SGLT2i but not ARNi, the majority (74%) received ACEi/ARBs to inhibit the renin–angiotensin system.

Therapeutic optimisation in HFrEF patients, facilitated by the integration of SGLT2i and the enhancement of other treatments, may elucidate two intriguing outcomes of our study. The first is related to the improvement in NYHA class and LVEF. Given the effective neurohormonal modulation and the not-yet-fully elucidated effects of SGLT2i [10–18], there is a high likelihood of improving both left ventricular systolic function and functional capacity. The second observation revolves around the administration of loop diuretics post SGLT2i. During follow-up, after the introduction of SGLT2i or further optimisation of disease modifiers' drug therapy, we tried to reduce the use of diuretics according to the clinical features of the patients, observing a significant reduction in the percentage of

patients receiving diuretic treatment. This could be attributable to the aforementioned improvements or the mild diuretic effects of both SGLT2i and sacubitril/valsartan [26,27]. The reduced need for diuretics may also have a beneficial pathophysiological effect by allowing the avoidance of adverse effects associated with loop diuretic use [28–30].

The final aspect for HFrEF patients pertains to therapy with beta-blockers and MRAs. For these two classes, a significant change in the prescription rate was not observed. However, as shown in Figure 2, there was a trend towards the use of higher doses, mirroring the pattern seen with sacubitril/valsartan. The capacity to introduce all four classes of medications and adjust their doses has profound clinical implications, as recently demonstrated in the STRONG-HF trial [20].

In our study, the prescription rate of SGLT2i for patients with HFmrEF and HFpEF was notably lower than anticipated. This was due to the fact that, up until June 2022 (which marked the end of the follow-up period), in Italy, the prescription of SGLT2i was reimbursed only for patients with T2DM. As a result, our efforts to initiate SGLT2i were focused primarily on T2DM patients with HFmrEF and HFpEF, which is reflected in the prescription rate and the prevalence of diabetic patients within the study. Despite this effort, the low percentage of HFmrEF/HFpEF patients prescribed SGLT2i underscores the significant lag of the Italian National Health System in aligning with the evidence from trials and the recommendations outlined in guidelines [4,15] and allowing the reimbursement of the new effective drugs. Such delays could lead to an elevated risk of HF progression, given the proven ability of SGLT2i to enhance prognosis irrespective of LVEF [5,6] and even in patients with HFimpEF [6].

The indications of the Italian National Health System for the reimbursement of the SGLT2i are also responsible for the disequilibrium in the prescription of dapagliflozin and empagliflozin. In fact, the reimbursement of dapagliflozin was allowed earlier than empagliflozin. Consequently, in most of our patients, dapagliflozin was prescribed, thus limiting the possibility of a comparison between the effects of the two drugs.

Limitations and perspectives. This observational study has several limitations. First, our study population had a notably high prevalence of males, which curtails the opportunity to discern gender-related differences among patients taking SGLT2i. Second, although we noted a trend toward increasing dosages of other disease-modifying drugs, a direct relationship with SGLT2i therapy cannot be ascertained based on our data alone. Moreover, the maximal dose of disease-modifying drugs achieved in our study was lower than that observed in the high-intensity care group of the recent STRONG-HF trial [12]. Future studies should elucidate these clinical nuances to fully understand how to optimise HFrEF therapy through the introduction and up-titration of the recommended drug classes. Moreover, the adverse events and safety profile should be adequately evaluated. Lastly, our ability to prescribe SGLT2i was limited to a small percentage of patients with HFmrEF, HFpEF, or HFimpEF. Real-world studies that explore the efficacy and effectiveness of SGLT2i across the LVEF spectrum would be valuable, especially considering the distinct pathophysiological bases and clinical determinants of HFmrEF and HFpEF [29].

## 5. Conclusions

In conclusion, our findings offer real-world evidence suggesting that SGLT2i therapy can be introduced to a significant proportion of patients with HFrEF. The introduction of this treatment does not hinder the optimisation of therapy with ARNi, beta-blockers, and MRAs. Moreover, it correlates with a decreased use of diuretics. Future research should validate these findings in a more extensive, multicentric real-world setting. Additionally, such studies should shed light on the clinical intricacies tied to the application of SGLT2i in patients with HFmrEF, HFpEF, or HFimpEF. It is worth noting that our results were constrained by Italian reimbursement policies.

**Author Contributions:** Conceptualization, E.T. and M.I.; methodology, M.C. and M.I.; formal analysis, M.I.; investigation, E.T., M.C., G.A., R.P., S.I., M.R., G.P. and M.I.; data curation, E.T., G.A., R.P., S.I., M.R., G.P. and M.I.; writing—original draft preparation, E.T. and M.I.; writing—review and editing, M.C., M.D.B. and N.D.B.; supervision, M.C., M.D.B. and N.D.B. All authors have read and agreed to the published version of the manuscript.

**Funding:** This research received no external funding.

**Institutional Review Board Statement:** The study was conducted in accordance with the Declaration of Helsinki, and approved by the Institutional Ethics Committee of the Polyclinic University Hospital of Foggia, Foggia, Italy (protocol code 68/CE/20, date of approval 26 May 2020).

**Informed Consent Statement:** Informed consent was obtained from all subjects involved in the study.

**Data Availability Statement:** The data are available on request.

**Conflicts of Interest:** The authors declare no conflict of interest.

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
