# Peer review of "Introduction of SGLT2 Inhibitors and Variations in Other Disease-Modifying Drugs in Heart Failure Patients: A Single-Centre Real-World Experience"

_clinpract, doi:10.3390/clinpract13050090_

Round 1

Reviewer 1 Report

1.     The quality of English editing must be reviewed : for example constant using the term “die“ instead of “day” – line 80-85.

Meanwhile, NYHA class in line 60 has been defined as II-II for study inclusion, most likely the authors referred to II-IV NYHA class.

2.     Regarding background, some information should be added, regarding the correlation between the pathophysiology and clinical effects of SGLT2i (ex: Palmiero G, Cesaro A, Vetrano E, Pafundi PC, Galiero R, Caturano A, Moscarella E, Gragnano F, Salvatore T, Rinaldi L, Calabrò P, Sasso FC. Impact of SGLT2 Inhibitors on Heart Failure: From Pathophysiology to Clinical Effects. Int J Mol Sci. 2021 May 30;22(11):5863. doi: 10.3390/ijms22115863. PMID: 34070765; PMCID: PMC8199383.).

3.     Regarding the results section, some data are confusing (from table 1):

a)     How can be explained the high percentage of male subjects -80% on inclusion -. This may affect the statistical evaluation since the adherence to some therapy – as betablockers - can be low?

b)     How can be explained the inclusion of patients with HF class I NYHA – 7.7% of the subjects -, since the indication is for II-IV NYHA patients?

c)     How can be explained the high percent of patients with HFmrEF and HFpEF with ICD or CRT on inclusion (35%)

d)     How can be explained the high percentage of patients with a low betablocker dose on inclusion (47.16% < 50% MRD). This aspect could influence the NYHA class on inclusion and also the loop diuretic dosage.

4.     In the discussion section we found no data regarding the influence on ACE inhibitors and ARB usage (figure 2), as 14% of patients were treated with ARB and 14% with ACE inhibitors on admission.

English Language can be reviewed (see above comments).

Author Response

We would like to thank the reviewer for his/her useful comments. This is our point to pint reply:

  1. The quality of English editing must be reviewed : for example constant using the term “die“ instead of “day” – line 80-85. Meanwhile, NYHA class in line 60 has been defined as II-II for study inclusion, most likely the authors referred to II-IV NYHA class.

Response:

The paper has been extensively reviewed by an English native speaker.

NYHA class refers to the Italian rules to make SGLT2i reimbursable.

  1. Regarding background, some information should be added, regarding the correlation between the pathophysiology and clinical effects of SGLT2i (ex: Palmiero G, Cesaro A, Vetrano E, Pafundi PC, Galiero R, Caturano A, Moscarella E, Gragnano F, Salvatore T, Rinaldi L, Calabrò P, Sasso FC. Impact of SGLT2 Inhibitors on Heart Failure: From Pathophysiology to Clinical Effects. Int J Mol Sci. 2021 May 30;22(11):5863. doi: 10.3390/ijms22115863. PMID: 34070765; PMCID: PMC8199383.).

Response:

We added a sentence in the introduction section and the suggested reference (#10).

  1. Regarding the results section, some data are confusing (from table 1):

  1. a) How can be explained the high percentage of male subjects -80% on inclusion -. This may affect the statistical evaluation since the adherence to some therapy – as betablockers - can be low?

Response:

The high percentage of male subject is frequently observed in observational study. We agree that this could represent a limitation of the study. For this reason, we added the following sentence in the discussion section at page 9, lines 281-283:

“Limitations and perspectives. This observational study has several limitations. First, our study population had a notably high prevalence of males, which curtails the opportunity to discern gender-related differences among patients taking SGLT2i.”

No gender dependent influence on beta-blocker therapy was observed.

  1. b) How can be explained the inclusion of patients with HF class I NYHA – 7.7% of the subjects -, since the indication is for II-IV NYHA patients?

Response:

We really thank the reviewer for the comment because we were probably not enough clear in describing our design of the study.

The aim was to understand in how many patients among all those referred to our heart failure unit it was possible to prescribe SGLT2i according with the current rules of Italian National Health System. Consequently, we considered also those patients in NYHA class I. This was one of the reason for the percentage that we observed in SGLT2i prescription (84%).

In order to better clarify this point we added in the methods section the following sentence at page 2, lines 71-72:

“For the study, all the patients with a history of CHF were considered, independently from LVEF, NYHA class and eligibility to SGLT2i therapy.”

  1. c) How can be explained the high percent of patients with HFmrEF and HFpEF with ICD or CRT on inclusion (35%).

Response:

This percentage is mainly related to the fact that these patients are those with heart failure and improved ejection fraction. We modified in to HFmrEF/HFpEF/HFimpEF.

We better clarified this point in the methods section at page 2, lines 89-92:

“Based on LVEF values, patients were classified as having HFrEF if their LVEF was <40%. The remaining patients were classified into the categories of HFmrEF, HFpEF, and HFimpEF in line with the current universal definition of HF [13].”

  1. d) How can be explained the high percentage of patients with a low betablocker dose on inclusion (47.16% < 50% MRD). This aspect could influence the NYHA class on inclusion and also the loop diuretic dosage.

Response:

The data about beta-blocker dose are in line with those reported in previous studies. In the trial SHIFT which was aimed to test ivabradine in patients with high heart rate and maximum tolerated dose of beta-blockers less than 50% of the patients were taking a beta-blocker dose =>50% of the maximum recommended dose.

In the more recent STRONG-HF, in the high intensity care group, a significantly greater percentage of patients taking half or more than half of disease modifiers drugs was reached.

We added this aspect as a limitation of the study at page 7 lines 283-287:

“Second, although we noted a trend towards increasing dosages of other disease-modifying drugs, a direct relationship with SGLT2i therapy cannot be ascertained based on our data alone. Moreover, the maximal dose of disease-modifying drugs achieved in our study was lower than that observed in the high-intensity care group of the recent STRONG-HF trial [12]. Future studies should elucidate these clinical nuances to fully understand how to optimise HFrEF therapy through the introduction and upward titration of recommended drug classes.”

  1. In the discussion section we found no data regarding the influence on ACE inhibitors and ARB usage (figure 2), as 14% of patients were treated with ARB and 14% with ACE inhibitors on admission.

Response:

We thank the reviewer, we added the following sentence at lines 241-243

“Finally, at the end of the follow-up, among the HFrEF patients taking SGLT2i but not ARNi, the majority (74%) received ACEi/ARBs to inhibit renin angiotensin system.”

Reviewer 2 Report

Dear Author(s), thank you for your interesting manuscript. Although well-written, all corrections according to the comments provided below should be made before potential publication.

-In the early introduction I would prefer that you state that SGLT2 should now be prescribed in HFmrEF/HFpEF patients (on top of HFrEF), regardless of diabetes. status ; and cite the relevant RCTs in this domain (efficacy in preserved EF setting)

-Table 1. Column HFrEF/HFpEF patients should be modified to HFmrEF/HFpEF patients?

-Also please explain why HFrEF group has 213 patients, and HFmrEF/HFpEF 137 patients, whilst the total numbers are 350 (your Table 1).

-Since you were determining interactions, it should also be nice if you could include AE rate / safety profile data.

-If I were you I would not mention canagliflozin in this setting, since its LV effectiveness is not yet determined as in dapa/empa RCTs (weighing of the evidence).

-In table 2 please separately add sections for HFrEF, HFmrEF, and HFpEF patients and present and compare all variables for each individual setting. Subanalyses should be performed and included within the main text or supplementary (decision up to the authors).

*Exclude canagliflozin patients from subanalyses

-Compare dapagliflozin and empagliflozin separately in two settings (or three in case of enough power/sample size) - HFrEF and HFmrEF/HFpEF.

-More limitations should be mentioned within the discussion section.

-Mentioned future directions for clinical and scientific setting.

-Include the comment mentioned in the discussion and cite Belančić&Klobučar, Diabetology, 2023, 4(3), 251-258 (https://doi.org/10.3390/diabetology4030022) : It would be useful in further randomized clinical studies and RWE studies to separately evaluate the efficacy/effectiveness of SGLT2 inhibition in patients with medium range and those with preserved ejection fraction, given the different pathophysiological bases and clinical determinants HFmrEF and HFpEF.

-Consider to make a graphical abstract.

Author Response

Dear Author(s), thank you for your interesting manuscript. Although well-written, all corrections according to the comments provided below should be made before potential publication.

Response:

We would like to thank the reviewer for his/her useful comments. This is our point to pint reply.

- In the early introduction I would prefer that you state that SGLT2 should now be prescribed in HFmrEF/HFpEF patients (on top of HFrEF), regardless of diabetes. status ; and cite the relevant RCTs in this domain (efficacy in preserved EF setting)

Response:

We modified the introduction as suggested. See page 1-2, lines 36-48:

“The beneficial effects across the entire spectrum of left ventricular ejection fraction (LVEF) are attributed to several hypothesised mechanisms that are not yet well clarified [10,11]. Certainly, the effects of SGLT2i in terms of reduction of glomerular hyperfiltration and preservation of glomerular filtration rate may play a pivotal role in cardiorenal protection [2, 11-13]. Beside these effects, other potential direct and indirect cardiac effects on cardiac function have been hypothesized, such as diuretic effect [14], improvement in myocardial energetics [15-16], reduction of cytosolic sodium and calcium levels and an increase in mitochondrial calcium [17]. Moreover, the increased cardiac delivery of oxygen due to the elevated hematocrit and the reduction of afterload could potentially enhance ventricular function [18]. On the basis of these hypotheses, it is likely to argue that these mechanisms are additive to those of the classes of drugs able to modulate neurohormonal activation [8, 10].”.

 -Table 1. Column HFrEF/HFpEF patients should be modified to HFmrEF/HFpEF patients?

Response:

We thank the reviewer for the suggestion. We corrected the mistake. Moreover, we reported that among patients with LVEF>40% a part was represented by patients with improved LVEF (HFimpEF).

 -Also please explain why HFrEF group has 213 patients, and HFmrEF/HFpEF 137 patients, whilst the total numbers are 350 (your Table 1).

Response:

The total number of patients enrolled was 350, among whom 213 showed a LVEF=<40% and the remaining 137 >40%.

 -Since you were determining interactions, it should also be nice if you could include AE rate / safety profile data.

Response:

We have only recorded the causes of SGLT2i withdrawal. We reported this as limitation of the study at page 9, line 284.

 -If I were you I would not mention canagliflozin in this setting, since its LV effectiveness is not yet determined as in dapa/empa RCTs (weighing of the evidence).

Response:

The two patients taking canaglifozin were affected by diabetic nephropathy. Both presented a LVEF>40%. According with these clinical features the diabetologists had prescribed canaglifozin before the enrollment in our study. We added this at page 4 lines 158-160.

“Notably, canagliflozin was the preferred choice by diabetologists before the study's onset for diabetes-afflicted patients with an LVEF >40%, coupled with compromised renal function and proteinuria.”

-In table 2 please separately add sections for HFrEF, HFmrEF, and HFpEF patients and present and compare all variables for each individual setting. Subanalyses should be performed and included within the main text or supplementary (decision up to the authors).

*Exclude canagliflozin patients from subanalyses

Response:

We added the table 1 the requested data.

-Compare dapagliflozin and empagliflozin separately in two settings (or three in case of enough power/sample size) - HFrEF and HFmrEF/HFpEF.

Response:

We added a table, which is the Table  in the current version, 2 which reports the requested data.

 -More limitations should be mentioned within the discussion section.

-Mentioned future directions for clinical and scientific setting.

Response:

We added a paragraph dedicated to limitations and perspectives at page 9, lines 281-294:

“Limitations and perspectives. This observational study has several limitations. First, our study population had a notably high prevalence of males, which curtails the op-portunity to discern gender-related differences among patients taking SGLT2i. Second, although we noted a trend towards increasing dosages of other disease-modifying drugs, a direct relationship with SGLT2i therapy cannot be ascertained based on our data alone. Moreover, the maximal dose of disease-modifying drugs achieved in our study was lower than that observed in the high-intensity care group of the recent STRONG-HF trial [12]. Future studies should elucidate these clinical nuances to fully understand how to optimise HFrEF therapy through the introduction and up-titration of the recommended drug classes. Moreover, the adverse events and safety profile should be adequately evaluated. Lastly, our ability to prescribe SGLT2i was limited to a small percentage of patients with HFmrEF, HFpEF, or HFimpEF. Real-world studies that explore the efficacy and effec-tiveness of SGLT2i across the LVEF spectrum would be valuable, especially considering the distinct pathophysiological bases and clinical determinants of HFmrEF and HFpEF.”.

-Include the comment mentioned in the discussion and cite Belančić&Klobučar, Diabetology, 2023, 4(3), 251-258 (https://doi.org/10.3390/diabetology4030022) : It would be useful in further randomized clinical studies and RWE studies to separately evaluate the efficacy/effectiveness of SGLT2 inhibition in patients with medium range and those with preserved ejection fraction, given the different pathophysiological bases and clinical determinants HFmrEF and HFpEF.

Response:

We added the comment in the discussion section, the paragraph dedicated to limitations and perspectives at page 9 lines 292-294. The reference was also cited as ref #29:

“Real-world studies that explore the efficacy and effectiveness of SGLT2i across the LVEF spectrum would be valuable, especially considering the distinct pathophysiological bases and clinical determinants of HFmrEF and HFpEF [29].”

 -Consider to make a graphical abstract.

Response:

We tried to do it.

Round 2

Reviewer 2 Report

All modifications are made as per comments.

Congratulations!